# The Effect of Co Content and Annealing Temperatures on the Resistivity in Ag-Co Films

**DOI:** 10.3390/nano12132297

**Published:** 2022-07-04

**Authors:** Yuanjiang Lv, Haoliang Sun, Pengyan Shi, Xinxin Lian, Haoge Zhang, Saibo Li, Shihao Liang, Guangxin Wang, Fei Ma

**Affiliations:** 1School of Materials Science and Engineering, Henan University of Science and Technology, Luoyang 471003, China; lyj_solovely@163.com (Y.L.); s18438603270@163.com (P.S.); lianxinxincn@163.com (X.L.); zhg70123@163.com (H.Z.); sur19980818@163.com (S.L.); l13253561073@163.com (S.L.); wgx58@126.com (G.W.); 2Henan Provincial Key Laboratory of Non-Ferrous Metal Material Science and Processing Technology, Luoyang 471003, China; 3School of Materials Science and Engineering, Xi’an Jiaotong University, Xi’an 710049, China; mafei@mail.xjtu.edu.cn

**Keywords:** film resistivity, Ag-Co films, annealing, ultra-high resistivity, negative ion trajectory

## Abstract

Ag-Co films with ultra-high resistivity were prepared on polyimide by magnetron sputtering. The effect of Co content and annealing temperatures on the resistivity and microstructure of Ag-Co films has been thoroughly investigated and the relation between resistivity and microstructure has been discussed. Results show that thicker Ag-Co films without annealing present lower resistivity due to better crystallinity. However, thin Ag-Co films (≤21 nm) annealed at 360 °C present ultra-high film resistivity because of the formation of diffusion pits on the film surface which blocks the transmission of electrons in films to increase film resistivity. Inversely, the resistivity of thick Ag-Co films (≥45 nm) annealed at 360 °C is much less than that annealed at lower than 260 °C owing to no diffusion pits. Furthermore, the addition of Co inhibits the growth of Ag grains and limits the migration of electrons in Ag-Co films further, also resulting in the increase of Ag-Co films’ resistivity.

## 1. Introduction

Electrical conductivities of metal or alloy films [1,2,3] have been widely studied by many researchers. Conductivities of films are often weaker than those of metal or alloy bulks [4,5] due to the presence of numerous point, line and surface defects in films. Therefore, many researchers [6,7,8] have done their utmost to improve film electrical conductivity for reducing the heat and energy loss generated by the working device to extend service life. The film devices with high electrical resistivity [1,2,9] have been also utilized in unique applications such as electrical heating devices and resistors in heating and cooling equipment and high-resistivity resistors working in high-voltage and high-temperature circumstances. There are many methods or treatment processes to increase electrical resistivity, for instance, alloy films [10,11] were oxidized to increase electrical resistivity owing to the wide band gaps and few free electrons of metal or alloy oxides [12,13].

Previous studies have focused on the electrical property [14,15] of granular films, aiming to enhance the conductivity of alloy films, and these granular films with an anomalous Hall coefficient could be a desirable and potential candidate for Hall devices. Many vacancies and dislocations [14] into metallic or alloyed films are present and seriously block electronic transport in these films to increase film electrical resistivity, which provides a feasible method to prepare high-resistivity films. A study about the local conductivity of lead zirconate titanate (PZT) films [16] revealed the mechanisms of current percolation determined by traps in the films and at grain interfaces. In another study, Boff et al. [17] showed that the electrical resistance of Co-Al_2_O_3_ granular films decreased with increasing volts in a low-field regime where granular films with a higher range of resistance variation were prepared. However, the preparation and treatment methods of the above granular films with high resistivity are complex.

In this work, based on previous studies about immiscible alloy films [18,19], the two immiscible metals of Ag and Co were sputtered and deposited on polyimide (PI) to prepare Ag-Co films under a stable magnetic field. The ratio of Ag and Co into granular films and film thicknesses would both affect film electrical resistivity, and hence the effects of Co content and film thicknesses on Ag-Co film conductivity were earnestly investigated. On the other hand, annealing temperature [14,15] is a crucial factor for the electrical resistivity variation of granular films; thus, we annealed these as-deposited Ag-Co films at different temperatures to examine the relationship between film resistivity and annealing temperatures. COMSOL Multiphysics software was employed to simulate the electron trajectory to explain and illustrate the critical influence of film surface morphology on film resistivity.

## 2. Materials and Methods

Ag-Co films were deposited on PI under a permanent magnetic field by sputtering a radio frequency composed target, which was made up of an Ag target (Φ 50 mm × 4 mm, 99.99% purity) and Co sheets (10 mm × 10 mm × 1 mm, 99.99% purity). After the background vacuum was down to 5 × 10^−4^ Pa, argon (Ar) was injected and the pressure and power of sputtering progression were set at 0.6 Pa and 100 W, respectively. Ag-Co films with different Co contents and film thicknesses were prepared by modifying the number of Co sheets and sputtering time at room temperature, respectively. Subsequently, the as-deposited Ag-Co films were annealed into a tube furnace, full of Ar, for 1 h. We used a blade to destroy the surface of the Ag-Co films and observed and measured the cross sections of the Ag-Co films. We calculated the average value of several times measured thicknesses and obtained the thickness of the Ag-Co films.

The sheet resistance of the Ag-Co films was measured by a four-probe resistance tester (RTS8, 4Probes Tech Ltd., Guangzhou, China) and the surface morphology of the Ag-Co films was observed by a field emission scanning electron microscope (FESEM, JSM-7800F, JEOL Ltd., Tokyo, Japan). In addition, the microstructure and composition of the Ag-Co films were obtained by high-resolution transmission electron microscopy (HRTEM, JEM-2100, JEOL Ltd., Tokyo, Japan) and energy-dispersive X-ray spectroscopy (EDS). The ion trajectory in the films was simulated by COMSOL Multiphysics software and the simulation with 1 μm × 1 μm × 0.2 μm was built in COMSOL Multiphysics software. Negative ions with 1 charge were injected to simulate the trajectory of electrons in the films.

## 3. Results and Discussion

### 3.1. Resistivity of Ag-Co Films with Different Thicknesses

The function of the sheet resistance (R_☐_) and the resistivity (ρ) of the films is shown in Equation (1):ρ = R_☐_ × d(1)
where d is the thickness of the Ag-Co films.

It can be observed from Figure 1 that the resistivity of the Ag-Co films reduces from 155.57 × 10^−6^ Ω cm to 39.15 × 10^−6^ Ω cm as the film thicknesses increase from 8 nm to 87 nm, in good agreement with the variation trend between film thicknesses and the resistivity of Cr-Si-Ta-Al films [2]. In another study about film thickness and crystallinity, C. Hsieh [20] provided evidence that thick films could improve the crystallinity of PbPc films. In terms of another study about film materials and film thicknesses by molecular dynamics, T. Yamamoto [21] further demonstrated that thick films present better crystallization than n-alkane ultrathin films. Thus, film resistivity reduces with the increase of Ag-Co film thickness due to better crystallization and fewer defects in Ag-Co films.

### 3.2. Resistivity of Ag-Co Films Annealed at Different Temperatures

The relationship between the film resistivity and annealing temperature of Ag-Co films is shown in Figure 2a. The Ag-Co films’ resistivity decreases by 44% with the increase of annealing temperatures from room temperature to 160 °C, in agreement with the results of Cr-Si-Ta-Al films [2], which indicates that high-temperature annealing temperatures could decrease film resistivity ascribed to improving film crystallization. However, Ag-Co film resistivity sharply increases or even cannot be measured when annealing temperature is hotter than 260 °C.

Figure 2b–e exhibit the surface morphology of the Ag-Co films annealed at different temperatures and the EDS result of the as-deposited Ag-Co films. There are no diffusion pits in the Ag-Co films annealed at lower than 160 °C in Figure 2c, and this film resistivity is lower than that of the as-deposited Ag-Co films. However, it can be seen from Figure 2d that many diffusion pits form on the surface of 21 nm Ag-Co films annealed at 260 °C and the resistivity of the Ag-Co films increases sharply due to the formation of diffusion pits blocking the in-film migration of electrons under the electric field. More seriously, the resistivity of the Ag-Co films annealed at 360 °C cannot be measured, ascribed to diffusion pits [22] amplifying, as shown in Figure 2e, to further limit the transmission of electrons.

### 3.3. The Analysis of Ultra-High Resistivity of Ag-Co Films

Ion trajectory could respond to the transport and migration direction of electrons, which could reveal the variation of resistivity in Ag-Co films. We employed COMSOL Multiphysics software to build the film simulation model with defects and calculate the negative ion trajectory in the films, as shown in Figure 3a–d.

Figure 3a shows the negative ion trajectory of films without any surface defects, and it can be found that the trajectory tracking presents straight lines, which indicates that films without any defects could display a low resistance to the transport of electrons. However, as surface defects form on films, the negative ion trajectories are deflected and the routes of electrons in films are increased, as shown in Figure 3b. Abundant diffusion pits on the film surface of Ag-Co films, as shown in Figure 2b–e, could increase the migration routes of electrons, which increases the resistivity of Ag-Co films. The routes of electrons in films become more curved and even broken as the surface defects become larger and increase in number, as shown in Figure 3c,d, which blocks and even cuts off the migration paths of electrons. Thus, the resistivity of Ag-Co films increases with the rise of annealing temperatures due to the formation of diffusion pits.

### 3.4. Resistivity of Annealed Ag-Co Films with Different Thicknesses

Figure 4a shows the resistivity of annealed Ag-18.9at% Co films with different thicknesses. It can be found from Figure 4a that the resistivity of Ag-18.9at% Co films decreases with the increase of film thicknesses after annealing at the same temperature, owing to thicker film thickness with better crystallinity [2,20]. As annealing temperatures increase from room temperature to 160 °C, the resistivity of Ag-18.9at% Co films decreases due to better crystallinity after annealing at lower than 160 °C. Furthermore, when annealing temperatures are lower than 160 °C, the resistivity of thin Ag-Co films (≤ 21 nm) is several times that of thick Ag-Co films (≥ 45 nm) due to the better crystallinity of thicker Ag-Co films. However, as annealing temperatures rise further, the resistivity of thin Ag-Co films sharply increases. It can be worth noting that the resistivity of thin Ag-Co films is more than 100 times that of thick Ag-Co films after being annealed at 260 °C, owing to the formation of numerous diffusion pits.

When Ag-Co film thickness is 8 nm, film resistivity increases with the increase of annealing temperature because the surface of thinner Ag-Co films is more sensitive to annealing temperature. Thus, the formation of many diffusion pits in the annealed films results in the decrease of Ag-Co film conductivity. When film thickness increases to 12 nm and 21 nm, film resistivity reduces first and then increases due to diffusion pits in Ag-Co films after annealing at higher than 260 °C. Alternatively, when the film thickness is more than 45 nm, the resistivity of Ag-Co films decreases with increasing annealing temperatures because of no diffusion pits and better crystallinity as shown in Figure 4b,c.

### 3.5. Resistivity of Ag-Co Films with Different Co Contents

It can be found that the resistivity of the Ag films is much higher than that of the theoretical value of 1.65 × 10^−6^ Ω·cm and that the resistivity of the Ag-Co films increases from 28.6 × 10^−6^ Ω cm to 134.45 × 10^−6^ Ω cm with the addition of more Co, as shown in Figure 5a. Obviously, there are many vacancies, dislocations and other kinds of defects in the Ag films, as shown in Figure 5c, to block electron migration, and therefore the resistivity of the Ag films is much higher than the theoretical. Moreover, the addition of Co in the films increases film resistivity and Ag-Co film resistivity increases with the increase of Co contents.

To understand the effect of Co contents on the resistivity of Ag alloy films, TEM was used to observe the microstructure of the Ag films and Ag-Co films. It can be clearly observed from Figure 5b–f that the grains are Ag grains, proven by the interplanar spacing in Figure 5(c1,e1,f1) and the fast Fourier transfer (FFT) images in Figure 5(c2,e2,f2), but no Co diffraction ring can be found in the FFT images of Figure 5e,f. Furthermore, amorphous matrices could be observed from the FFT images in Figure 5(e3,f3), which are deduced as amorphous Co matrices. It is worth noting that the size of the Ag grains in the Ag-Co films is much smaller than that in the Ag films, which implies the addition of Co inhibits the growth of Ag grains (compare Figure 5b,d). Furthermore, more defects and grain boundaries are formed in Ag-Co films, as shown in Figure 5d–f, to block the transmission of electrons to increase the resistivity of Ag-Co films.

## 4. Conclusions

The high resistivity of Ag-Co films is related to film thicknesses, annealing temperatures and Co contents. It was found that the Ag-Co film resistivity decreases with the increase of film thicknesses due to the better crystallinity of thicker Ag-Co films. Furthermore, the resistivity of Ag-Co films increases with the increase of Co contents due to the block effect of longer grain boundaries and defects. When annealing temperatures are lower than 160 °C, the resistivity of thin Ag-Co films (≤21 nm) is several times that of thick Ag-Co films (≥45 nm). Alternatively, when annealing temperatures are more than 160 °C, the resistivity of 45 nm Ag-Co films is less than 1/100 times that of 21 nm Ag-Co films due to no diffusion pits and better crystallinity in Ag-Co films. Thin Ag-Co films with ultra-high film resistivity may become a potential candidate for high-resistance devices.

## Figures and Tables

**Figure 1 nanomaterials-12-02297-f001:**
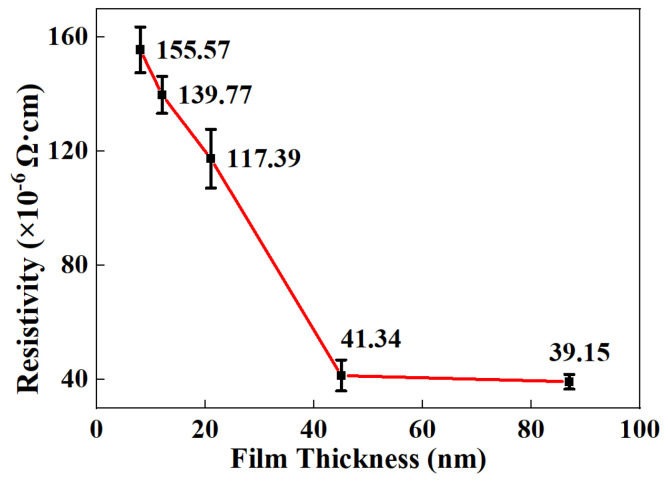
The variation of film resistivity of as-deposited Ag-18.9at% Co films with thicknesses ranging from 8 nm to 87 nm.

**Figure 2 nanomaterials-12-02297-f002:**
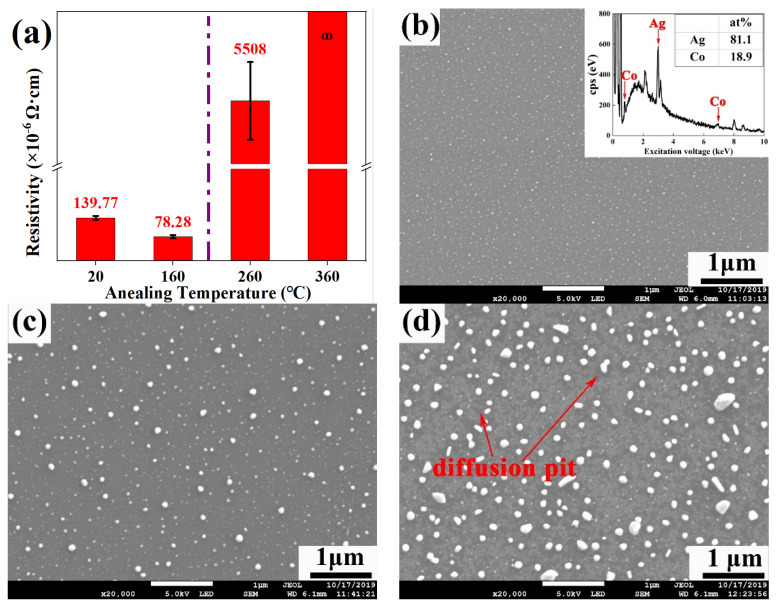
(**a**) Resistivity variation of 21 nm Ag-Co films annealed from room temperature to 360 °C; (**b**) surface morphology and EDS pattern of as-deposited 21 nm Ag-Co films; (**c**–**e**) surface morphology of 21 nm of Ag-Co films annealed at different temperatures, respectively: (**c**) 160 °C, (**d**) 260 °C and (**e**) 360 °C.

**Figure 3 nanomaterials-12-02297-f003:**
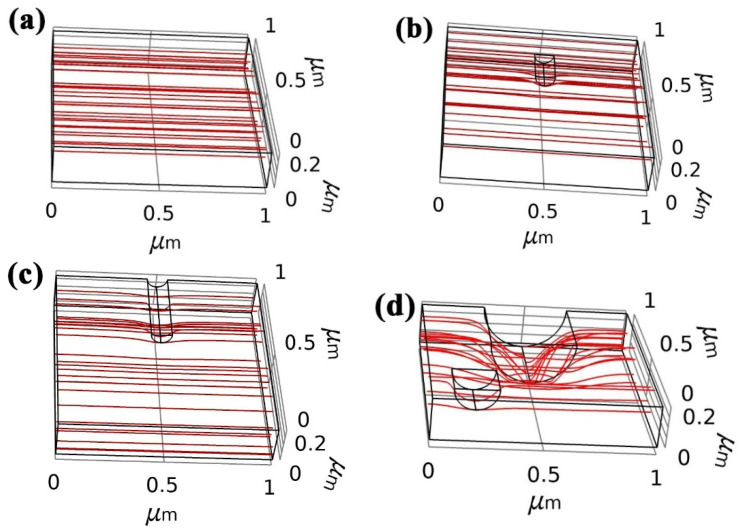
(**a**–**d**) The simulation results of negative ion trajectory in films with different-size surface defects calculated by COMSOL Multiphysics software: (**a**) no surface defects, (**b**) φ 0.05 × 0.2 μm column surface defect, (**c**) φ 0.05 × 0.4 μm surface defects and (**d**) φ 0.2 × 0.4 μm and φ 0.2 × 0.4 μm surface defects.

**Figure 4 nanomaterials-12-02297-f004:**
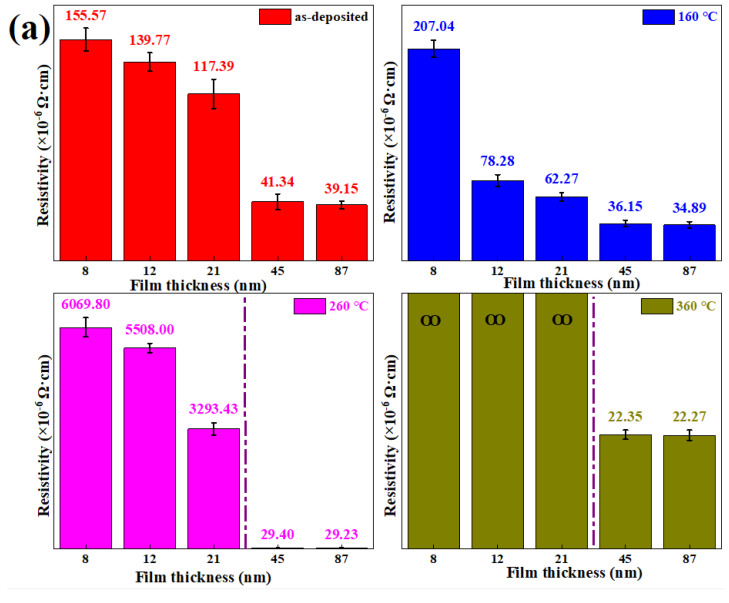
(**a**) The resistivity of annealed Ag-18.9at% Co films with film thicknesses ranging from 8 nm to 87 nm; (**b**,**c**) surface morphology of 45 nm and 87 nm Ag-18.9at% Co films annealed at 360 °C, respectively.

**Figure 5 nanomaterials-12-02297-f005:**
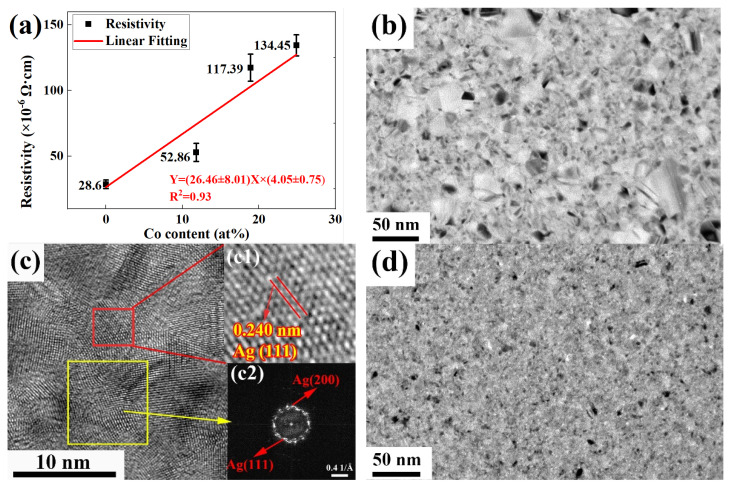
(**a**) The variation of the resistivity of Ag-Co films with Co contents ranging from 0at% to 24.9at%; (**b**) the low magnification TEM image of Ag film; (**c**,**c1**,**c2**) HRTEM image, enlarged image of the red rectangle region and Fast Fourier transformation (FFT) image of the yellow rectangle region Ag films, respectively; (**d**) the low magnification TEM image of Ag-18.9at% Co film; (**e**,**e1**,**e2**,**e3**) HRTEM image, enlarged image and FFT image of the red rectangle region and FFT image of the yellow rectangle region Ag-18.9at% Co films, respectively; (**f**,**f1**,**f2**,**f3**) HRTEM image, enlarged image and FFT image of the red rectangle region and FFT image of the yellow rectangle region Ag-24.9at% Co films, respectively.

## Data Availability

The data that support the findings of this study are available from the corresponding authors on reasonable request.

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
