# Peer review of "The Effect of Co Content and Annealing Temperatures on the Resistivity in Ag-Co Films"

_nanomaterials, 2022, doi:10.3390/nano12132297_

Round 1

Reviewer 1 Report

The experimental work done in Lv et.al are very well and the results are very good, however the authors have not done enough justice to the work in the write up. Before going for the publication, author should address some major comments on this work.

 Comment #1: The ultra-high resistive samples are prepared and characterized very well. It has nice findings. However, the title of the manuscript and the abstract should be revised.

Author should change the title of the manuscript, something like “the effect of Co content in Ag-Co films and temperature in the resistivity. The title should be catchy; not exactly, what I said above but should contains some findings. The author should rewrite the abstract for better readable.

Comment#2: The insert figure in Fig 2b are not very clearly visible. The figure captions does not contains enough details. Please add few more lines in the captions, which can explains figures more details.

Comments#3: All the figures in the manuscript should be explained precisely. Results and discussion are not written in details. Author should do proper justice to the nice experimental results by writing and explaining them in details.

Comments#4: The author showed nice HRTEM image in the figure 5, however, they should explain them carefully. HRTEM images gives here only local information. Unless, they show a low magnification TEM image, it is very difficult to tell about the quality of the films. I would appreciate, if they provide a low magnification TEM images in the manuscript. 

Reviewer 2 Report

The presented manuscript concerns studies on the structural and electrical characteristics of Ag-Co films with ultra-high resistivity. The thin films were deposited through magnetron sputtering by using a Ag target and Co sheets. Then, the samples were annealed at 160, 260 and 360 C. The samples were characterized before and after the thermal annealing treatment in terms of electrical and structural properties and results were correlated with the resulted thicknesses. The topic is fully consistent with the aims and objectives of the journal. However, I cannot recommend this manuscript for publication in its present form. Thus, my suggestion is major revision. The following are my comments and suggestions.

-       The introduction part should be expanded

-       What was exactly the diameter of Ag target, 50 or 50.8 mm?

-       The magnetic field was originating only from the magnets situated beneath the target or it an additional magnetic field was used? Besides, it is not clear if all deposition runs were made in Direct Current Magnetron Sputtering mode.

-       The function of sheet resistance might be moved from Materials and methods to Results and discussion part.

-       How the thicknesses were measured? Please specify the type and dimensions of the used substrates.

-       I cannot understand the connection between the Ag – Co films and Cr-Si-Ta-Al, PbPc or n-alkane ultrathin films. Please discuss.

-       Please modify the figures legend. Figures 1, 2, 5 have an opening such as “the relation between..”

-       The Comsol Multiphysics software is not presented in Materials and methods part.

-       Please insert information between every single SEM image in order to include magnification, accelerating voltage etc.

-        Do you consider that the appearance of the diffusion pits can be triggered by the surface morphology of the used substrates?

Conclusively, the study is not ready for publication.

Round 2

Reviewer 2 Report

Authors have modified the manuscript carefully. Just one step is needed before acceptance:

-       Please insert in the manuscript the information revealed on the previous response regarding the measured thicknesses “we used a blade to destroy the surface of Ag-Co film and observed the cross sections of Ag-Co films under FESEM. We calculated the average value of several times measured thicknesses and got the thickness of Ag-Co films.”
